# Wear and Corrosion Resistance Analysis of FeCoNiTiAl$_x$ High-Entropy Alloy Coatings Prepared by Laser Cladding

## Zhaolei Sun, Mingyuan Zhang, Gaoqi Wang, Xuefeng Yang and Shouren Wang *

School of Mechanical Engineering, University of Jinan, Jinan 250022, China; s2431310571@163.com (Z.S.); me_zhangmy@ujn.edu.cn (M.Z.); me_wanggq@ujn.edu.cn (G.W.); me_yangxf@ujn.edu.cn (X.Y.)
* Correspondence: me_wangsr@ujn.edu.cn; Tel.: +86-531-82765476

**Abstract:** FeCoNiTiAl$_x$ ($x$ = 0, 0.5, 1) high-entropy alloy coatings were prepared by laser cladding technology. The phase, microstructure, hardness, wear resistance and corrosion resistance were tested and analyzed. The results showed Al element promoted the conversion from the FCC phase to the BCC phase. The coating presented dendritic structure due to the addition of the Al element, while the number of dendrites increased. And the average hardness of the coating increased from 204 to 623 HV. The addition of the Al element increases the corrosion current density of the coating from $1.270 \times 10^{-5}$ to $3.489 \times 10^{-5}$ A/cm$^2$. The wear rate of the coatings decreases with the increase of Al content according to dry friction and wear, which indicates the wear resistance of the coating was improved by adding the Al element. According to the corrosion wear test in 3.5% NaCl solution, it can be found that the wear rate of the coating increases firstly and then decreases with the addition of the Al element, which indicates that the addition of the Al element intensifies the wear of the coating in 3.5% NaCl solution.

**Keywords:** high-entropy alloy coating; laser cladding; dry friction and wear; corrosion friction and wear

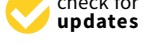



## 1. Introduction

The concept of the high-entropy alloys (HEAs) was proposed in 1995. Due to the unique composition of HEA, it has attracted much attention. Generally speaking, HEAs are usually composed of five or more elements, and the atomic percentage of each element in the composition is 5% to 35% [1,2]. It has been developed for nearly 20 years through the unique concept of composition design and has become a new metal material with excellent properties [3,4]. Compared with traditional alloys, the high-entropy alloys have four unique effects, namely high entropy effect, delayed diffusion effect, lattice distortion effect and "cocktail" effect [5]. Due to the interaction of these four effects, the high-entropy alloy has a high mixing entropy value, which reduces the Gibbs free energy and makes the alloy system tend to be stable, thus showing excellent microstructure and mechanical properties [6,7]. Recently, excellent resistance, corrosion resistance, abrasion resistance, high-temperature resistance, and oxidation resistance have become important research hotspots in the field of metal materials [8–11]. Because HEAs have high mixing entropy, it is easier to form simple solid solution phases, such as body-centered cubic phase (BCC) and face-centered cubic phase (FCC), rather than complex intermetallic compound phases [12,13]. Therefore, the main strengthening mechanism of high-entropy alloys is solid solution strengthening [14,15]. Each multi-component alloy system can be designed as a simple equal atomic molar ratio alloy or a non-equal atomic molar ratio alloy, and minor elements can be added to improve the performance of the alloy. By adding Al, Ti, Mo and other elements with larger atomic radius to the coating, the hardness and wear resistance of the coating will be significantly improved. Xu et al. [16] prepared CoCrFeNiTiAl$_x$ high-entropy alloy coating by laser cladding. The results showed that the addition of Al improves the hardness and wear resistance of the coating, and at the same time improves

the corrosion resistance in 3.5% NaCl solution. Guo et al. [17] studied $CoCr_2FeNiTi_x$ high entropy alloy coatings. It was found that the addition of Ti atoms can significantly improve the hardness and tribology properties of the coating. Besides, literature [18] reported that the Mo element has a great influence on the mechanical properties of the coating. In the $Al_2CrFeNiMo_x$ high entropy alloy coating, as the Mo content increases, the hardness of the coating also increases, and the maximum hardness was three times that of the stainless steel substrate.

As a kind of material with low price and good mechanical properties, Q235 steel is widely used in industrial production. However, due to the high humidity and high salinity of seawater, Q235 steel is very susceptible to corrosion in the marine environment, which destroys the original structure of the material and causes the material to fail. Therefore, the preparation of coatings on the metal surface can solve the questioning problem well, and it has received extensive attention. At present, the main methods for preparing high-entropy alloy coatings include hot-press sintering [19–22], laser cladding [23,24], and magnetron sputtering [25]. Laser cladding is a new method for preparing high entropy alloy coatings that have been developed in recent years. The advantages of this method lie in the high density and high energy of the laser, rapid heating, and rapid cooling. Different laser power has different effects on the morphology and performance of the coating. Wang et al. [26] found that when the laser power is 3000 W, the mechanical properties of the FeCoCrNiMo coating reach the best.

In order to improve the wear resistance and corrosion resistance of Q235 steel in seawater, Q235 steel is selected as the substrate, and the high-entropy alloy coating is prepared on the surface of the substrate by laser cladding technology. In this paper, $FeCoNiTiAl_x$ ($x$ = 0, 0.5, 1) high-entropy alloy coatings were prepared by laser cladding technology. A scanning electron microscope (SEM) was used to analyze the microstructure of the coating, and X-ray diffraction (XRD) was used to analyze the phase composition of the coating. A friction and wear tester was used to test the wear resistance of the coating. A self-made corrosion and wear tester was used to study the wear rate and wear scar depth of the coating after corrosion wear in 3.5% NaCl solution.

## 2. Materials and Methods

### 2.1. Preparation of Experimental Materials

Fe, Co, Ni, Ti, Al powder (purity > 99.5%, particle size 75~105 μm) was selected as the original powder, and the $FeCoNiTiAl_x$ ($x$ = 0, 0.5, 1) alloy system was weighed with an electronic balance. The subscript of the system indicates the atomic percentage among each element, and the default value is 1 where there is no subscript. The specific content of each element is shown in Table 1. The $FeCoNiTiAl_x$ ($x$ = 0, 0.5, 1) high-entropy alloy powders can be prepared by ball milling. The mass ratio of the grinding ball to the alloy powder is 5:1, the rotation speed of the ball mill is set to 220 r/min, and the milling time is 2 h. After the ball milling, put the ball-milled alloy powder into a drying box and dry it at 80 °C for 5 h to ensure the fluidity and dryness of the powder. In order to improve the wear resistance and corrosion resistance of Q235 steel in seawater, Q235 steel was selected as the substrate, the composition of Q235 steel was shown in Table 2. The synchronous powder feeding laser cladding equipment (LAM400s, leiShi, Jinan, China) was used to prepare high-entropy alloy coatings. The working principle of laser cladding is shown in Figure 1. The laser cladding parameters were shown in Table 3.

**Table 1.** The content of each alloy element.

| Alloy | Fe (at.%) | Co (at.%) | Ni (at.%) | Al (at.%) | Ti (at.%) |
|---|---|---|---|---|---|
| FeCoNiTi | 25.0 | 25.0 | 25.0 | 0 | 25.0 |
| FeCoNiTiAl0.5 | 22.1 | 22.2 | 22.3 | 11.2 | 22.2 |
| FeCoNiTiAl | 20.0 | 20.0 | 20.0 | 20.0 | 20.0 |

**Table 2.** The composition of Q235 steel (wt.%).

| Elements | C | Si | Mn | S | P | Fe |
|----------|------|------|------|------|------|------|
| Content | 0.16 | 0.14 | 0.46 | 0.02 | 0.02 | 99.2 |

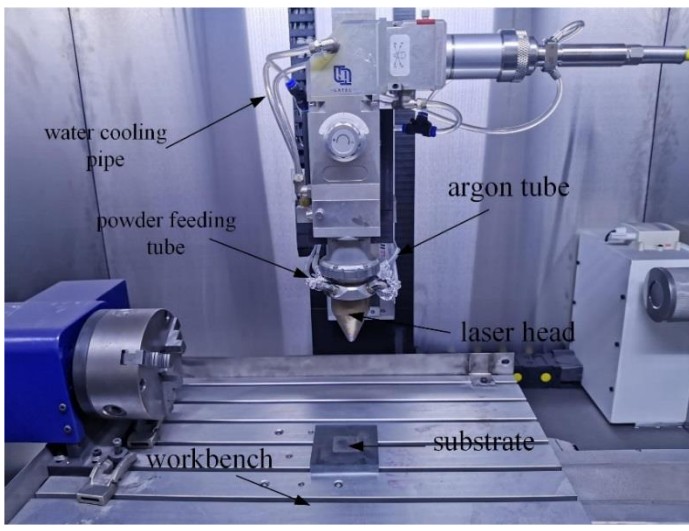

**Figure 1.** Laser cladding equipment.

**Table 3.** The parameters of laser cladding.

| Laser Power | Scanning Rate | Spot Shape | Spot Diameter | Overlap Rate |
|-------------|---------------|------------|---------------|--------------|
| 1.2 KW | 300 mm/min | round | 2 mm | 40% |

*2.2. Experimental Methods*

For the coated substrate, the sample was cut into 10 mm × 10 mm samples by a wire cutting machine, and the sandpaper was used for stepwise grinding. Then, the samples were mechanically polished with a polishing agent. Finally, the polished sample was cleaned with alcohol by ultrasonic wave, and the cleaned coating samples were placed in a drying oven and dried at 80 °C for 2 h.

A scanning electron microscope (SEM) (JSM-7610F, Japanese electronics, JEOL, Tokyo, Japan) was used to observe the microstructure of the FeCoNiTiAl$_x$ coating. At the same time, the energy dispersive spectroscopy (EDS) (X-max, Oxford, UK) was used to perform the element distribution of the coating cross section. By X-ray diffraction (XRD) (D8-ADVANCE, Bruker, Karlsruhe, Germany), the phase composition of the FeCoNiTiAl$_x$ coating was detected with a Cu target. The parameter conditions of X-ray diffraction were as follows: the voltage was 36 KV, the current was 20 mA, scanning speed was 4°/min, test range was 20°~90°. The Vickers hardness tester (402MVD, Wilson, NC, USA) was used to measure the step hardness from the surface of the FeCoNiTiAl$_x$ coating to the base material. During measuring, recording the hardness value every 60 μm along the section direction with the applied load of 500 g and the retention time of 15 s. An electrochemical workstation (chi604e, Chenhua Equipment, Shanghai, China) was used to analyze the corrosion resistance of the FeCoNiTiAl$_x$ coating. In the experiment, a three-electrode method was used for electrochemical testing. The sample was used as the working electrode, the calomel electrode was used as the reference electrode, and the platinum electrode was used as the auxiliary electrode. Electrochemical experiments were carried out in 3.5% NaCl solution with the scanning potential of −1.5~−0.2 V.

The friction and wear tester (RTEC, MFT-50, San Jose, CA, USA) was used to perform dry friction and wear tests on the coating. The test parameters were shown in Table 4. A white light interferometer was used to measure the wear volume and wear depth of the

coating after dry friction and wear. Figure 2 shows the friction and wear tester. Corrosion wear requires simultaneous measurement of the electrochemical corrosion and tribology behavior of the coating. Corrosion wear equipment was shown in Figure 3. In the corrosion and friction test, the sample was used as the working electrode, saturated KCl was used as a reference electrode, and a platinum electrode was used as a counter electrode. The test parameters were shown in Table 5. By the white light interferometer, the wear rate and wear depth of the coating after corrosion friction and wear were measured. In the friction and wear experiment, the material of the grinding ball was GCr15, and the composition of GCr15 was shown in Table 6.

**Table 4.** The parameters of dry friction and wear.

| Normal Load | Friction Method | Frequency | Friction Time | Ball Material |
|---|---|---|---|---|
| 50 N | reciprocating | 4 Hz | 30 min | GCr15 |

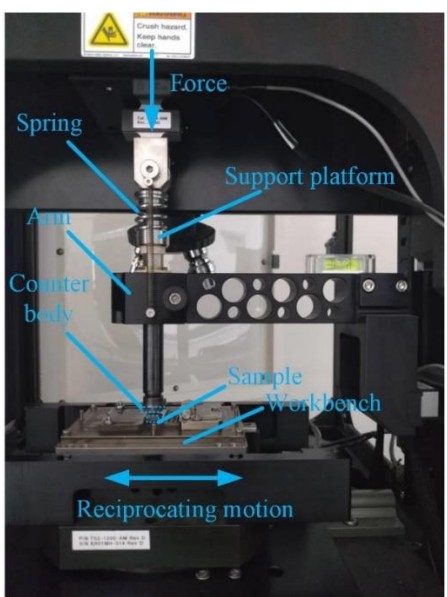

**Figure 2.** Friction and wear test device.

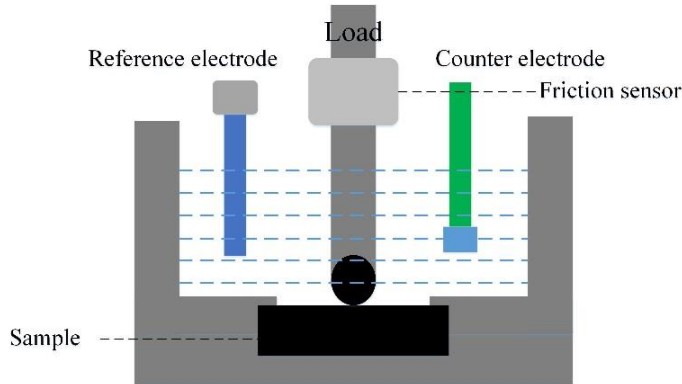

**Figure 3.** Corrosion friction and wear test device.

**Table 5.** The parameters of corrosion friction and wear.

| Normal Load | Friction Method | Frequency | Friction Time | Ball Material |
|---|---|---|---|---|
| 50 N | reciprocating | 4 Hz | 70 min | GCr15 |

**Table 6.** The composition of GCr15 (wt.%).

| Elements | C | Si | Mn | Cr | P | Fe |
|---|---|---|---|---|---|---|
| Content | 0.95 | 0.25 | 0.35 | 1.50 | 0.02 | 96.93 |

## 3. Results and Discussion

### 3.1. Phase Composition of Coatings

Figure 4 shows the XRD pattern of FeCoNiTiAl$_x$ high-entropy alloy coatings prepared by laser cladding. The results show that HEA coating is mainly composed of a solid solution with a simple structure. The main phase of Al$_0$ coating is the FCC phase. With the addition of the Al element, the phase structure of the Al$_{0.5}$ coating has changed a lot, mainly composed of BCC crystal structures. When Al = 1, the phase composition of Al$_1$ coating is similar to Al$_{0.5}$·coating. It can be seen from the enlarged image the BCC diffraction peak is slightly shifted to the left. The reason is that the atomic radius of the Al element is larger, and the solid solution of the Al element will cause lattice distortion and expansion, and the lattice constant will increase, so the diffraction peak is slightly shifted to the left.

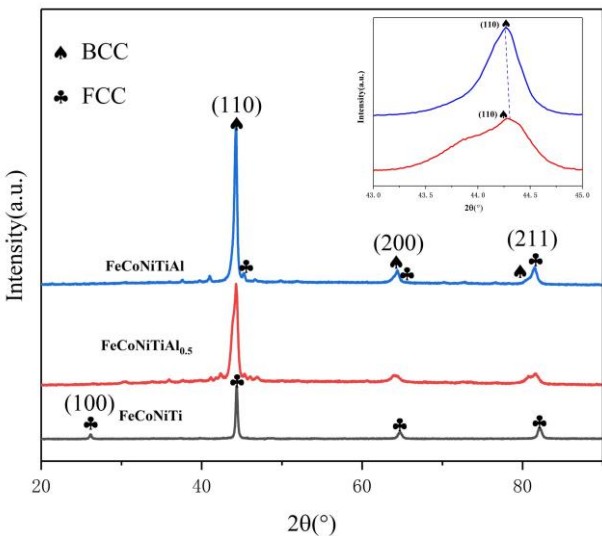

**Figure 4.** The XRD patterns of the FeCoNiTiAl$_x$ high-entropy alloy.

The results of the XRD analysis show that the addition of the Al element has a significant effect on the phase composition of the FeCoNiTiAl$_x$ high-entropy alloy coating. Since the density of the FCC crystal structure is higher than that of the BCC crystal structure, the addition of the Al element causes the lattice distortion and the lattice distortion energy of the original FCC crystal structure to increase. When the energy increases to the point that the FCC crystal structure cannot be maintained continuously, the metastable FCC phase tends to transform into the relatively stable BCC phase in order to maintain the energy balance of the system.

### 3.2. Microstructure of Coatings

Figure 5 shows the microstructure morphology of FeCoNiTiAl$_x$ high-entropy alloy coating. Defects such as "black spots" can be found on the surface of the coating. With the increase of Al content, the microstructure of FeCoNiTiAl$_x$ ($x$ = 0~1) high-entropy alloy coating changed significantly. The Al$_0$ alloy coating is basically all gray (Figure 5a). At this composition ratio, the main phase of the coating is a uniform FCC single-phase structure. When Al = 0.5, Al$_{0.5}$ coating presents a typical a dendritic structure, composed of dendritic zone (DR) and an interdendritic zone (ID). As the content of Al continues to increase, the morphology of the structure does not change significantly. Compared with Al$_{0.5}$ (Figure 5(b-1)), the dendrite size of Al1 alloy coating Figure 5(c-1) is basically

the same, but the number of dendrites is larger and the distance between dendrites is smaller. According to XRD pattern (Figure 4), the main phase of FeCoNiTiAl$_x$ ($x$ = 0.5, 1) high-entropy alloy coating is BCC phase. Therefore, we can infer that BCC is concentrated in the dendrite area according to (Figure 5(b-1,c-1)). FeCoNiTiAl coating was selected for EDS analysis, and the result is shown in Figure 5d. We can see that region 1 is enriched with Ni, Al and Ti elements, and the content of the Fe element in region 2 is relatively high, the distribution of Co element is more uniform. Due to the different affinity between elements, element segregation is prone to occur. The segregation of elements is related to the mixing enthalpies between elements. The smaller the enthalpy of mixing, the greater the affinity between the two elements [16]. According to Table 7, the mixing enthalpy between the Fe element and the other four elements is high, so the affinity is relatively weak. In contrast, the enthalpy of mixing among the three elements of Ni, Ti, and Al elements is low, so they have a strong affinity and are finally enriched in region 1.

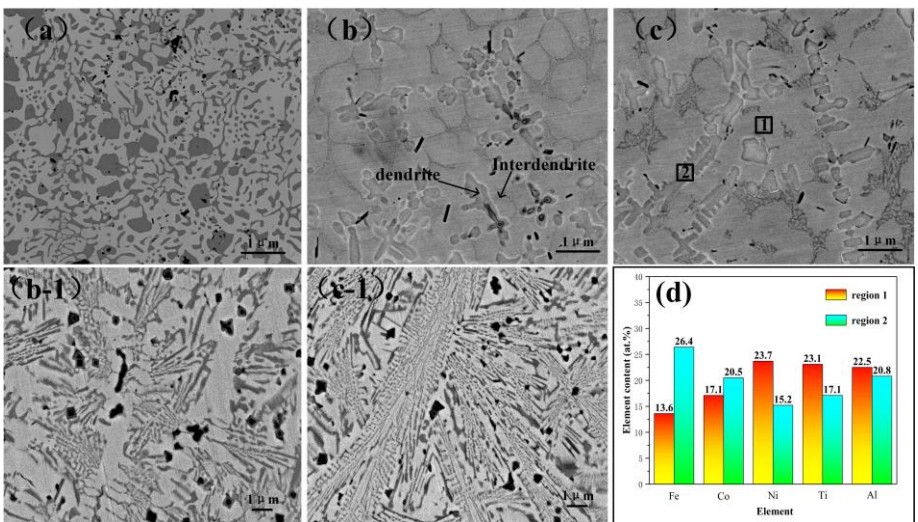

**Figure 5.** The microstructure of FeCoNiTiAl$_x$ coatings and the EDS analysis (**d**) of FeCoNiTiAl coating. (**a**) FeCoNiTi-3000X; (**b**) FeCoNiTiAl$_{0.5}$-3000X; (**c**) FeCoNiTiAl-3000X; (**b-1**) FeCoNiTiAl$_{0.5}$-5000X; (**c-1**) FeCoNiTiAl-5000X.

**Table 7.** Mixing enthalpies between elements (kJ mol$^{-1}$).

| Elements | Fe | Co | Ni | Al | Ti |
|----------|------|------|------|------|------|
| Fe | — | −1 | −2 | −11 | −17 |
| Co | −1 | — | 0 | −19 | −28 |
| Ni | −2 | 0 | — | −22 | −35 |
| Al | −11 | −19 | −22 | — | −30 |
| Ti | −17 | −28 | −35 | −30 | — |

Figure 6 shows the SEM picture of the Al$_1$ high-entropy alloy coating cross-section perpendicular to the single-pass cladding direction. It can be seen that the coating and the substrate are metallurgically bonded, and the bonding is tight and continuous. It can be seen from the cross-sectional line scan results that the content of the Fe element drops sharply from a high value at the interface, and finally drops to a lower value. The other four metal elements are completely contrary to the law of the Fe element, they increase sharply near the interface, and the change range is smaller than the Fe element. In the process of laser cladding, the mixed metal powder is melted by the laser beam into the molten pool. At high temperatures, all the metal elements will diffuse into the substrate, and the Fe element in the substrate will also diffuse into the coating, which increases the Fe content in the coating.

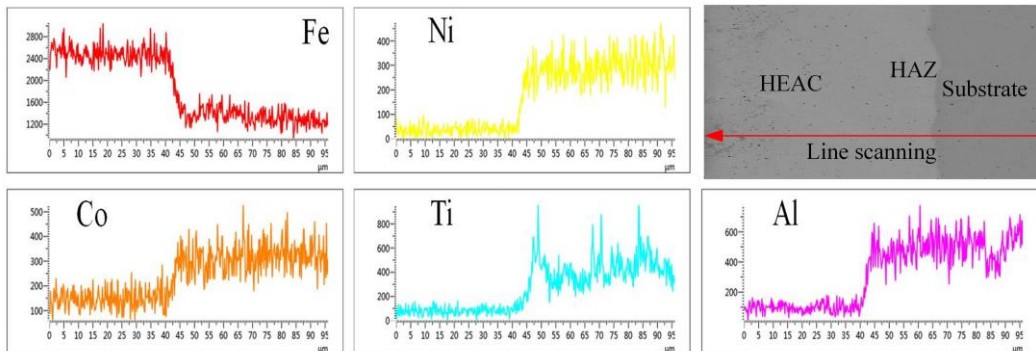

**Figure 6.** The EDS line scan pattern of the FeCoNiTiAl coating section.

### 3.3. Microhardness of Coatings

Figure 7 shows the microhardness distribution curve of the FeCoNiTiAl$_x$ high-entropy alloy coating section. It can be seen that the hardness distribution curve is in a stepped shape, and the curve is divided into three parts, corresponding to the cladding zone, heat-affected zone and substrate. From the surface cladding layer to the substrate, the hardness distribution shows a decreasing trend, and the decreasing interval is concentrated in the heat-affected zone. It can be seen from the figure that the hardness of the high-entropy alloy coating is significantly higher than that of the substrate. The highest average hardness of the Al$_1$ coating (623 HV) is approximately 3.1 times the average hardness of the substrate (204 HV).

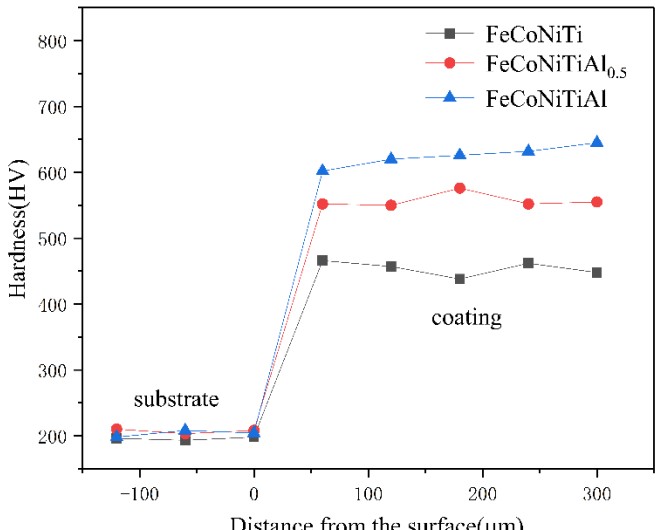

**Figure 7.** Microhardness of FeCoNiTiAl$_x$ high-entropy alloy coatings section.

The high hardness of high-entropy alloys is attributed to the following three main reasons: First, during the laser cladding process, rapid heating and rapid cooling make the grains of the microstructure refined and increase the solubility. According to Figure 5, it can be seen that the addition of the Al element makes the number of dendrite structures increase and the distance between dendrites decreases, which achieves the effect of fine grain strengthening. Second, the difference in atomic size leads to lattice distortion and solid solution effects. The atomic radius of the Al element is much larger than that of other elements, which increases lattice distortion and the solid solution effect. Third, the BCC structure has a higher hardness than the FCC structure, and the addition of the Al element promotes the formation of the BCC phase.

### 3.4. Corrosion Resistance of Coatings

Figure 8 shows the polarization curve of FeCoNiTiAl$_x$ high-entropy alloy coatings in 3.5% NaCl solution, and the corrosion parameters of coatings are as shown in Table 8. It can be seen from Table 7 that with the increase of Al content, the corrosion current density of FeCoNiTiAl$_x$ high-entropy alloy coating in 3.5% NaCl solution gradually increases. The self-corrosion potential of the Al$_1$ coating is the most positive ($-0.867$ V), but the self-corrosion current density is the highest ($3.489 \times 10^{-5}$ A/cm$^2$).

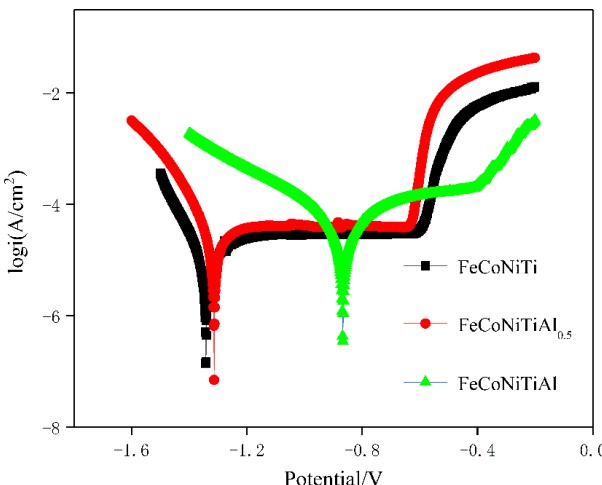

**Figure 8.** Polarization curves of FeCoNiTiAl$_x$ high-entropy alloy coatings in 3.5% NaCl solution.

**Table 8.** Corrosion parameters of FeCoNiTiAl$_x$ high-entropy alloy coatings in 3.5% NaCl solution.

| Alloy | Ecorr/(V) | Icorr/(A/cm$^2$) |
|---|---|---|
| FeCoNiTi | $-1.343$ | $1.270 \times 10^{-5}$ A/cm$^2$ |
| FeCoNiTiAl0.5 | $-1.314$ | $2.704 \times 10^{-5}$ A/cm$^2$ |
| FeCoNiTiAl | $-0.867$ | $3.489 \times 10^{-5}$ A/cm$^2$ |

Al$_0$ coating is the main FCC phase, the structure is more uniform, and the corrosion resistance is better. Because Al is a passivating element, the addition of a small amount of Al element will cause a passivation film to be formed on the surface of the alloy when the alloy is corroded, thereby playing a role in protecting the substrate. Compared with Al$_0$ coating, the microstructure of Al$_{0.5}$ coating has changed and dendrites appear. At the same time, the grain boundaries of the coating have increased significantly. Therefore, when the applied potential moves towards the anode, the grain boundaries have higher dislocation density and corrosion activity. So they are more susceptible to corrosion damage, which will promote the corrosion of the coating, and cause the corrosion current density to increase [16]. With the further increase of Al element, the self-corrosion potential of Al$_1$ coating is obviously shifted to the anode direction than that of Al$_{0.5}$ coating, indicating that its corrosion resistance has improved a lot. The introduction of the Al element promotes the passivation behavior of the coating to form passivation film. The passivation film will prevent the corrosion of the coating by Cl ions in the 3.5% NaCl solution.

### 3.5. Dry Friction and Wear

Figure 9 shows the friction coefficient of the FeCoNiTiAl$_x$ high-entropy alloy coating and substrate under dry friction and wear conditions. It can be seen that the friction coefficient of the substrate is greater than that of the coating under the same parameters, which is about 0.452. When $x = 0$, the sliding friction coefficient of the alloy fluctuates around 0.408, which is a relatively high level overall. When the Al content increases to 0.5, the sliding friction coefficient decreases sharply and drops to about 0.349. As the Al

content in the alloy coating continues to increase, the sliding friction coefficient continues to decrease, but the decrease is significantly lower than the former, and the average friction coefficient of $Al_1$ coating fluctuates around 0.334. Generally speaking, the sliding friction coefficient can also reflect the wear resistance of the material to a certain extent. The smaller the sliding friction coefficient, the smaller the horizontal resistance of the material in the sliding process. The material is not easy to be damaged, relatively speaking, the coating material has better wear resistance.

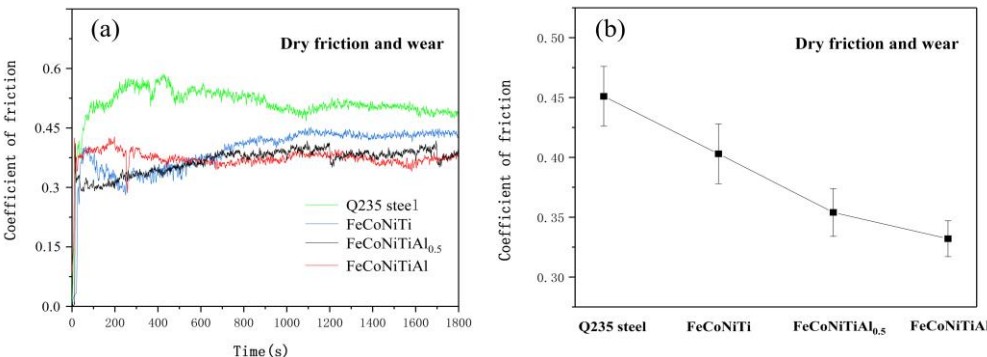

**Figure 9.** The friction coefficient curve (**a**) and value (**b**) of FeCoNiTiAl$_x$ coating and substrate under dry friction and wear.

It can be seen from Figure 10a that the wear volume of the substrate reached the maximum, and it is 4.7 mm$^3$. Due to the low hardness of the substrate, the surface of the substrate cannot withstand the continuous action of a tangential force. Under the action of normal load and constant reciprocating friction, a large amount of material falls off, which increases the wear volume and the depth of the wear scar (Figure 10b). In addition, the wear volume of the coating decreases with the increase of Al content, which is consistent with the friction coefficient of coating (Figure 9). When $x = 0$, the wear volume is 3.5 mm$^3$. As the Al content increases, the wear volume of the coating continues to decrease, and the wear volume of the Al$_{0.5}$ alloy coating drops to 2.8 mm$^3$. When Al = 1, the wear volume reaches the minimum value of 2.5 mm$^3$. At the same time, it can be seen from Figure 10f that with the addition of the Al element, the depth of the coating wear scar gradually decreases from 20 to 16 μm. Besides, the inner side of the coating wear scar gradually becomes smooth from Figure 10c–e. The addition of the Al element improves the hardness and wear resistance of the coating, so that the coating is not easily damaged when subjected to tangential and normal loads, thereby making the inside of the wear scar gradually smooth.

The worn surface morphology of FeCoNiTiAl$_x$ high-entropy alloy coating under dry friction and wear is shown in Figure 11. It can be seen from Figure 11a that many grooves and a lot of material peeling are found on the surface of Al$_0$ coating. In contrast, due to the high hardness and strength of the Al$_{0.5}$ coating material, the grooves formed on the worn surface are very shallow, and the pits caused by the material peeling are reduced. With the increase of Al content, the worn surface morphology of Al$_1$ coating is improved. The number of furrows increases, but the depth decreases. At the same time, the large amount of material peeling off decreases. There are only a small number of pits, and plastic deformation occurs. At this time, the wear mechanism of the coating is slight abrasive wear and accompanied by oxidative wear.

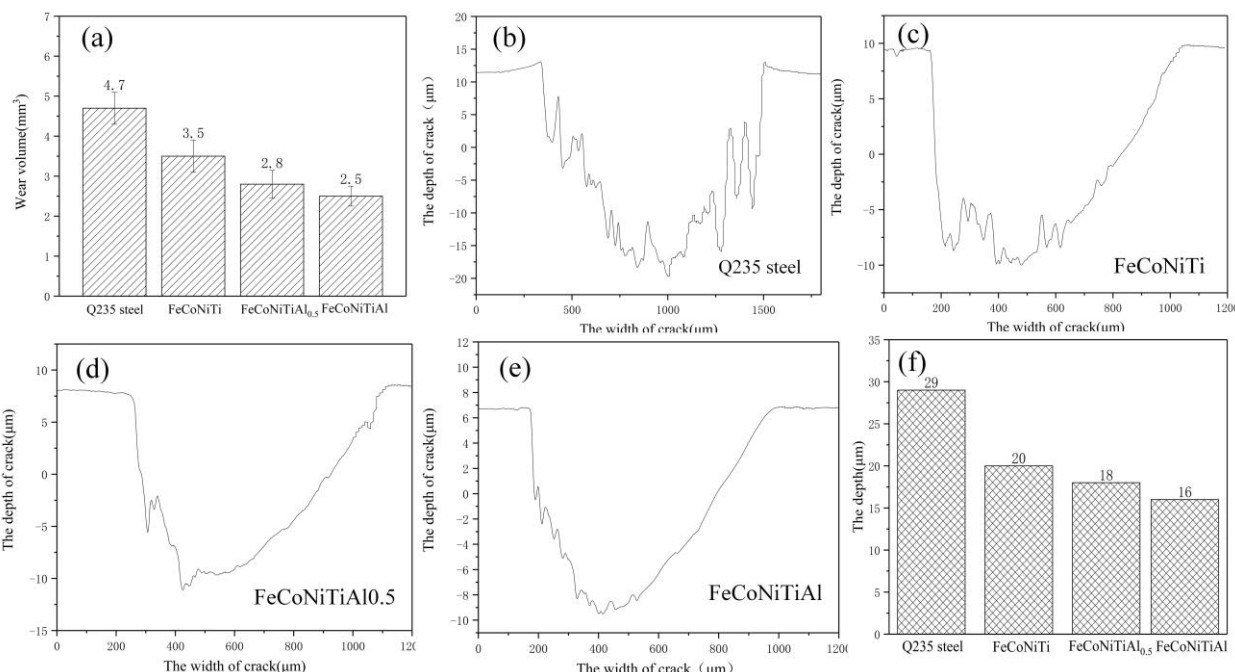

**Figure 10.** The wear volume (**a**) and wear depth (**b**–**f**) of FeCoNiTiAl$_x$ coating and substrate under dry friction and wear.

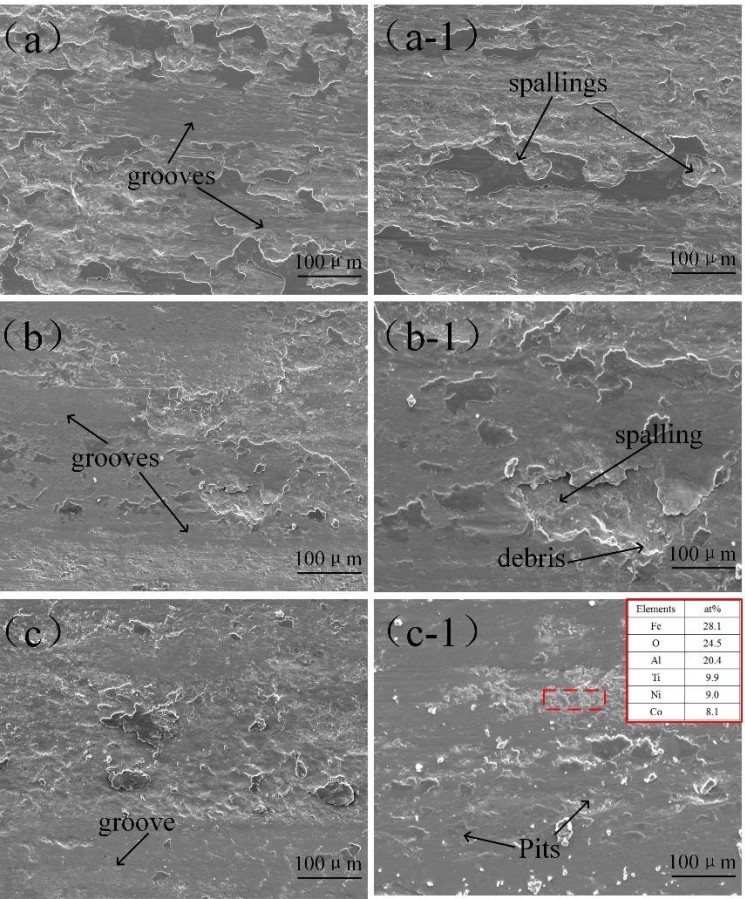

**Figure 11.** SEM images of the worn surface of FeCoNiTiAl$_x$ high-entropy alloy coating under dry friction and wear. (**a**) FeCoNiTi-200X; (**b**) FeCoNiTiAl$_{0.5}$-200X; (**c**) FeCoNiTiAl-200X; (**a-1**) FeCoNiTi-500X; (**b-1**) FeCoNiTiAl$_{0.5}$-500X; (**c-1**) FeCoNiTiAl-500X.

Due to the relatively low hardness of $Al_0$ coating, fatigue cracks are formed under the continuous action of normal load and tangential force, which will eventually lead to a large amount of material peeling. Therefore, the wear mechanism of $Al_0$ coating is abrasive wear and fatigue wear. Under dry friction and wear, it is the direct contact between the counter body and the coating surface, and the normal load and tangential shear applied that make the temperature of the counter body and the coating surface increase. The Al element is relatively active and easily oxidizes at high temperatures to form an oxide film, thereby improving the surface morphology of the wear scar and reducing the large-area peeling phenomenon of the coating surface.

*3.6. Corrosion Friction and Wear*

Figure 12 shows friction coefficient of the $FeCoNiTiAl_x$ high-entropy alloy coating and substrate under corrosion friction and wear. Compared with the dry friction results, the friction coefficient of the coatings and substrate in 3.5% NaCl solution decreases, indicating that the NaCl solution can indeed play a certain lubricating effect. The friction coefficient of the substrate in the 3.5% NaCl solution is the highest, which is 0.357. Fe is the main component of the substrate. It is easily corroded under the corrosive environment of 3.5% NaCl solution, which intensifies friction and wear and leads to an increase in friction coefficient. The curve of $Al_0$ coating is relatively flat during the whole friction process, and its friction coefficient is at least 0.258. With the addition of the Al element, the curve appears to fluctuate. The fluctuation of $Al_{0.5}$ coating is the most obvious, and its average friction coefficient is up to 0.302. Compared with $Al_{0.5}$, the friction coefficient curve of $Al_1$ coating tends to be stable, and the average friction coefficient is 0.283.

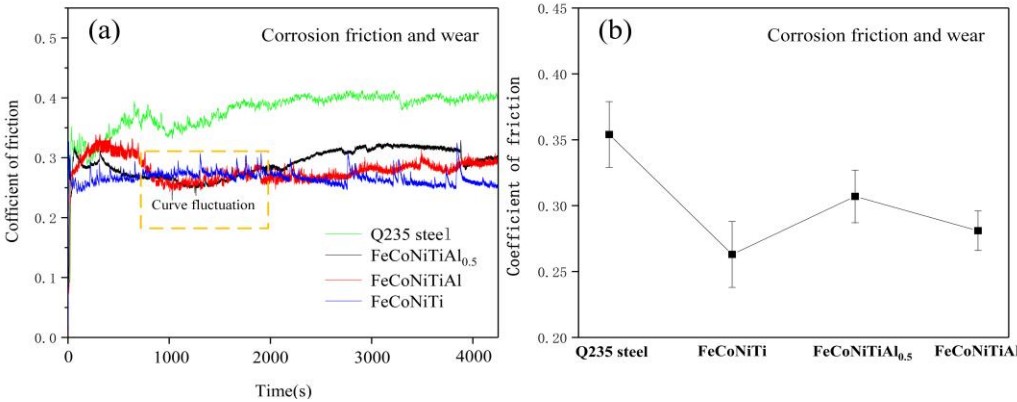

**Figure 12.** The friction coefficient curve (**a**) and value (**b**) of $FeCoNiTiAl_x$ coating and substrate under corrosion friction and wear.

It can be seen from Figure 13a the wear rate of the substrate reaches the highest value ($8.8 \times 10^{-5}$ mm³/N·m). And the wear rate of the coating is at least $6.1 \times 10^{-5}$ mm³/N·m without Al element. With the addition of the Al element, the wear rate of the coating increases. When Al = 0.5, the wear rate of the coating reaches $7.5 \times 10^{-5}$ mm³/N·m. The wear rate ($6.8 \times 10^{-5}$ mm³/N·m) of $Al_1$ coating is lower than that of $Al_{0.5}$ coating, but it is still higher than that of $Al_0$ coating. According to Figure 13f, it can be seen that the wear depth of the substrate is greater than that of the $FeCoNiTiAl_x$ coating, with a maximum value of 18 μm. The addition of the Al element makes the wear depth increases firstly, and then decreases under corrosion friction and wear, and the wear depth of FeCoNiTi coating reaches the lowest (9 μm). Besides, it can be seen from Figure 13c to 13e that the addition of the Al element makes the contour curve of the wear scar change from smooth to rough. There are two main reasons for this phenomenon: firstly, friction and wear directly lead to the pits caused by the peeling of the coating, and secondly, the pores caused by the corrosion of the active metal.

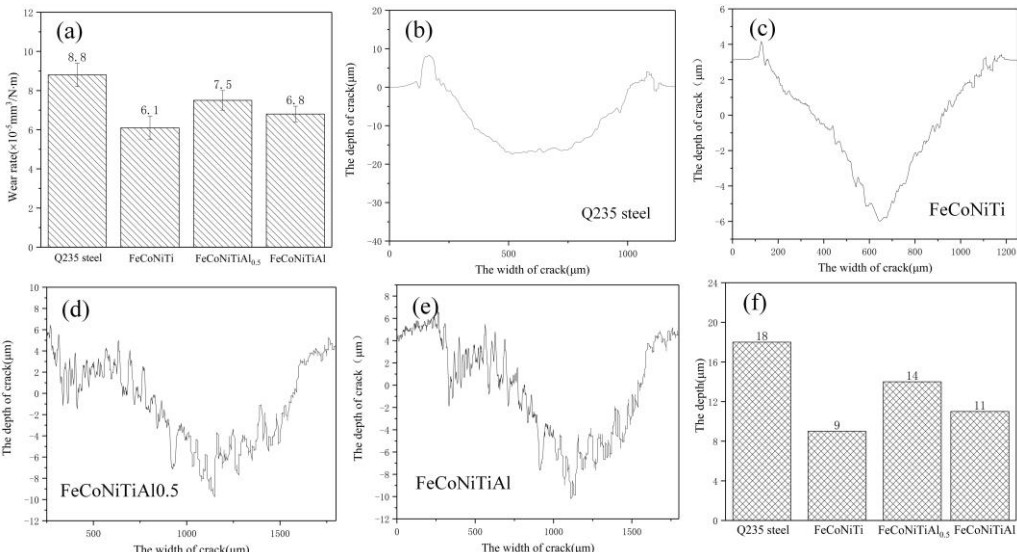

**Figure 13.** The wear rate (**a**) and wear depth (**b**–**f**) of FeCoNiTiAl$_x$ coating and substrate under corrosion friction and wear.

It can be seen from the experiment that the wear resistance of the coating is better than that of the substrate regardless of dry friction and wear or corrosion friction and wear. The experimental results show that the addition of the Al element improves the hardness and wear resistance of the coating. Under dry friction conditions, the Al$_1$ coating has the lowest wear rate and exhibits good wear resistance. In the 3.5% NaCl solution, the wear rate of Al$_0$ coating is the lowest, and that of Al$_{0.5}$ coating is the highest for FeCoNiTiAl$_x$ high-entropy alloy coating. Under the condition of applied potential, the corrosion and passivation rate of the FeCoNiTiAl$_{0.5}$ coating surface is faster, and rapid passivation and continuous wear occur simultaneously during the corrosion friction and wear process. The passivation film is continuously destroyed and removed during the friction process. This vicious circle causes the corrosion area to increase and the wear volume increases. With the increase of Al element content, the passivation film produced by corrosion increases, which relieves the corrosion effect to a certain extent, thereby reducing the wear rate. In general, the addition of a small amount of Al element will cause the corrosion wear resistance of the coating to decrease under corrosion friction and wear. At the same time, the corrosion will aggravate the wear of the coating, which leads to an increase in the wear rate.

The worn surface morphology of FeCoNiTiAl$_x$ high-entropy alloy coating under corrosion friction and wear is shown in Figure 14. It can be seen from Figure 14a that a large number of grooves appear on the coating surface, and small pieces of material on the coating surface peel off around the grooves. This may be because the material on the surface is destroyed and the granular debris is not discharged in time, which plows the surface of the wear scar. It shows that the wear mechanism of the Al$_0$ coating is mainly abrasive wear and corrosion wear. With the addition of the Al element, the number of grooves on the coating surface decreases. This is because the addition of the Al element increases the hardness of the coating. When Al = 0.5, a large number of corrosion pits appear on the surface of the coating, and fatigue cracks appear at the same time. According to Figure 5(b-1), the addition of the Al element makes the dendritic structure obvious, and the area where the corrosive liquid enters the grain boundary increases, resulting in obvious corrosion. Under the action of long-term normal load, wear disrupts the passivation film and thus increases the corrosion effect. Meanwhile, fatigue cracks appeared on the surface of the coating. Cracks will continue to enlarge under the action of liquid corrosion and tangential cutting force, eventually forming pits. It can be found from Figure 14c that the grooves on the worn surface of the Al$_1$ coating are further shallower, which is related to the highest hardness and best wear resistance of the Al$_1$ coating. The corrosion current density of the Al$_1$ coating is the highest. Therefore, when fatigue cracks appear on the surface of wear marks, corrosion will aggravate wear and lead to the appearance of spalling.

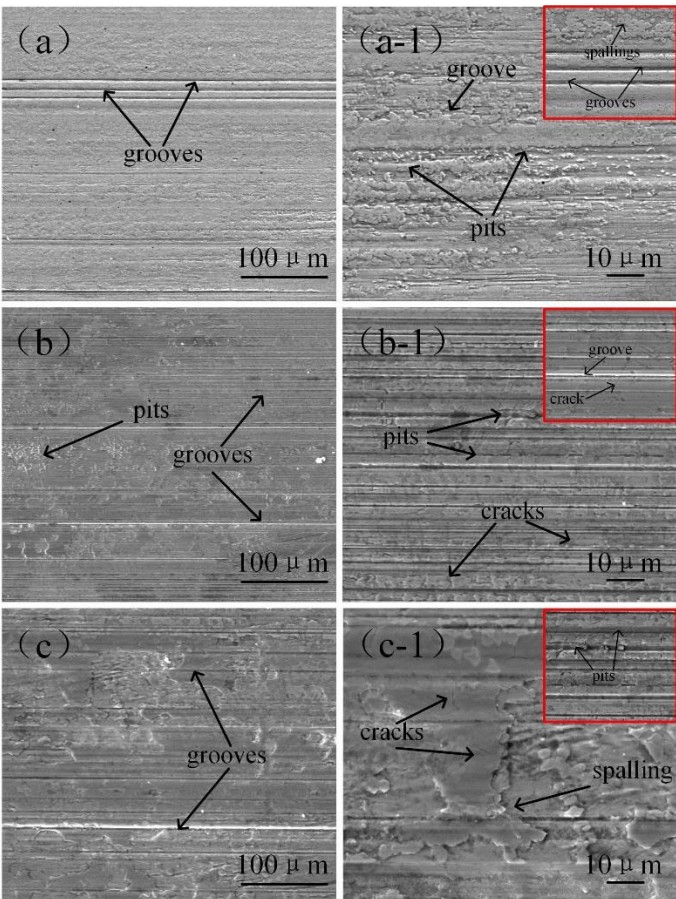

**Figure 14.** SEM images of the worn surface of FeCoNiTiAl$_x$ high-entropy alloy coating under corrosion friction and wear. (**a**) FeCoNiTi-500X; (**b**) FeCoNiTiAl$_{0.5}$-500X; (**c**) FeCoNiTiAl-500X; (**a-1**) FeCoNiTi-2500X; (**b-1**) FeCoNiTiAl$_{0.5}$-2500X; (**c-1**) FeCoNiTiAl-2500X.

Compared with the dry friction and wear, the wear mechanism of the coating under the corrosion friction and wear has not changed much, and it is still the mixed wear of abrasive wear and fatigue wear. However, the coating will be corroded in 3.5% NaCl solution, which will aggravate wear. Because the liquid has a certain lubricating effect, the abrasive wear is weakened and the corrosion wear is aggravated.

## 4. Conclusions

In this study, FeCoNiTiAl$_x$ ($x$ = 0, 0.5, 1) coatings were synthesized by laser cladding. The phase and microstructure of the coatings were analyzed by XRD and SEM, and the corrosion resistance and wear resistance of the coatings were investigated by electrochemical experiments and sliding wear tests. The following conclusions can be summarized:

(1) The main phase of FeCoNiTi coating is the FCC crystal structure. With the increase of Al content, the phase composition of the coating has changed. The addition of the Al element promotes the phase transformation from the FCC phase to the BCC phase. At the same time, the microstructure presents a typical dendritic structure because of the addition of the Al element.

(2) The hardness of FeCoNiTiAl$_x$ ($x$ = 0,0.5,1) coating increases with the addition of the Al element, and the hardness of FeCoNiTiAl coating reaches the highest, it's 623 HV. The introduction of the Al element promotes the self-corrosion potential to move towards the anode direction from −1.343 to −0.867 V, but increases the corrosion current density from $1.270 \times 10^{-5}$ to $3.489 \times 10^{-5}$ (A/cm$^2$).

(3) The addition of the Al element reduces the friction coefficient and wear volume of the coating, the morphology of the worn surface is modified at the same time. So the

FeCoNiTiAl$_x$ coatings exhibit better wear resistance than that of the substrate under dry friction and wear.

(4) Compared with dry friction and wear, the wear rate of the coating increases firstly and then decreases with the addition of the Al element, which indicates that the addition of the Al element intensifies the wear of the coating in 3.5% NaCl solution.

**Author Contributions:** Investigation, Data curation, and Writing—original draft preparation, Z.S.; Validation, Resources, Project administration and Funding acquisition, S.W.; Investigation, Writing— review and editing M.Z.; Writing—review and editing, G.W.; Formal analysis, Supervision, X.Y. All authors have read and agreed to the published version of the manuscript.

**Funding:** This work was supported by the National Natural Science Foundation of China (No. 51872122), Major basic research projects of Shandong Natural Science Foundation (ZR2020ZD06), the Independent Innovation Team Funding in Jinan (No. 2019GXRC012), Key projects of Shandong Natural Science Foundation (ZR2020KE062) and Taishan Scholar Engineering Special Funding (No. ts201511040).

**Institutional Review Board Statement:** Not applicable.

**Informed Consent Statement:** Not applicable.

**Data Availability Statement:** Not applicable.

**Conflicts of Interest:** The authors declare no conflict of interest.

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
