# Peer review of "Wear and Corrosion Resistance Analysis of FeCoNiTiAlx High-Entropy Alloy Coatings Prepared by Laser Cladding"

_coatings, doi:10.3390/coatings11020155_

Round 1

Reviewer 1 Report

Evaluation of the paper “Wear and Corrosion Resistance Analysis of FeCoNiTiAlx High Entropy Alloy Coatings Prepared by Laser Cladding”

The beginning of abstract is OK yet the final part is very confusion and I suggest reformulation; especially the part “According to corrosion friction and wear, the wear rate and wear depth of the coating increased due to the addition of Al element, indicating that the corrosion wear resistance of the coating was weakened. Besides, the wear rate of coatings decreases with the increase of Al content according to dry friction and wear, which indicates the wear resistance of coating was improved by adding Al element.”

The introduction starts with a very confusion phrase, in which the English is very weak..so ,I suggest  to check the entire work for English formulation

“ The concept of the high-entropy alloys (HEAs) was proposed the concept in 1995” I am not sure if the concept propose the concept !!!

“ The HEAs are composed of five or more elements” OK but there are plenty HEA which one do you refer cause then you indicate only 2

You discuss about HEA alloys and give details of TiN particles!!!

Which is the industrial impact of HEA alloys and in which components do you use them ?

“The cost of preparing high-entropy alloy bulk materials is relatively high, and the preparation of coatings can meet the needs of materials in terms of performance, so it has attracted widespread attention.” Seems that do not have logic if they are high the industry may avoid them…

“Different elements have different effects” this is obvious …

There is difficult to understand the novelty …

In conclusion for introduction it should be reformulated and presented in a better manner.

“was selected as the research object”?? rather was selected for manufacturing the HEA in this research

“Put the weighed alloy powder into the QE-04 planetary ball mill and mix it evenly.” I suspect that the authors refer to the manufacturing process ..so please redraft the materials and methods section

“Q235 steel was selected” why this was selected and not another one ???

Please be more clear about this “Polishing cloth and diamond polishing” cause is difficult to understand which diamond size was used

“Table 2.The” between . and T requires a space …please check entire paper for typo mistake too

This “50N “ has any physical meaning…The same for other parameters in Table 1-4!!

Fig 4, 6,9, 10, 13 quality are very poor

You have a section of results and discussion. The results are somehow very interesting but the discussion part is missed please elaborate and discuss your results against literature

Author Response

Thank you very much for your opinion, it will provide important guidance for my future scientific research work.

Reviewer 2 Report

In this paper, the authors analyzed wear and corrosion resistance of FeCoNiTiAlx (x = 0,0.5,1) high-entropy alloy coatings were prepared by laser cladding technology.
The topic is suitable for “Coatings” journal.
My opinion is that the authors obtained many results which may be proved useful for practical application.

Author Response

(The authors gave the same response as above.)

Reviewer 3 Report

Authors of the paper are estimating compositions of multicomponent alloys under study as high entropy alloys (HEA). It is supposed that HEA are forming uniform sold solutions to get maximal entropy of mixing.

For alloys that are discussed in the paper one can clearly see on microstructure photos (Fig. 5 (a - c)) that there are forming 3 phases (dark grey dendrites, light grey interdendrite phase and black particles) with clearly visible interfaces and different compositions regardless of rapid crystallysation at laser cladding. So these phases should be identified by their compositions and lattice parameters because this information is critically important for further discussion of properties. Quantities of phases should be calculated.  This information could be obtained from TEM+EDS studies or some other combination of methods with high spatial resolution.

I can guess from microstructures, diffraction data and text that most probably the dendrite and inter-dendrite phases are fcc ones with different lattice constants while black particles have bcc lattice.

On my mind, the observed increase in hardness, corrosion behaviour and wear resistance of the coatings should be associated with refining microstructure and growing amount of black particles.

Corrosion and wear behaviour of the coatings under study should be compared with uncoated steel in the same conditions.

So the paper can be published only after the set of additional experiments.

In addition to these key flaws, the paper in its present form requires serious corrections.

  1. Please use SI units: for example, powder particles size should be listed in micrometers rather in mesh (line 73).
  2. Q235 and GCr15 steels. Please add references to corresponding Chinese standards in English.
  3. Table 1: there is obvious error in this line regarding Al and Ti content. You do not need also to repeat "at.%" together with the elements names: it was already declared in the table caption.
  4. Please provide Fig. 4 enlarged and with better resolution. Please add also plains indices of plains to corresponding diffraction peaks.
  5. Microstructure photos (Figs. 5) should be enlarged to resolve better details of the alloys' structure. There is some discordance in Figs (b) and (c) with scales on main microstructures and on zoomed inserts. 
  6. You mention presence of some amount bcc phase in the alloy without Al (line 201) but you mention in discussion of Fig. 4 that only fcc phase was found in this alloy. 
  7. Line 213: Presence of alumina and other protective oxides (chromium oxide, for example) on the samples' surface should be approved experimentally.

Author Response

(The authors gave the same response as above.)

Round 2

Reviewer 1 Report

.

Author Response

Thank you very much for your comment on my article, it will be of great help to my future scientific research work.

Reviewer 3 Report

This version of the paper was significantly improved comparing to its initial state and could be published in Metals after minor edition by authors. Please enter proper subscript/superscript formatting through the text: for example, use FeCoNiTiAlx instead of FeCoNiTiAlx (line 9). Numbers with powers of 10 should be also corrected: please use 1.270⋅10-5 A/cm2  (line 15) instead of 1.270e-005 A/cm2.

I still insist that authors have hurried with publication leaving unattended the most interesting questions about compositions and and lattice parameters of phases (dendrites, inter-dendrite space, black particles) forming during rapid crystallisation of multi-element coatings. In fact they got not classical uniform high entropy alloy but some mixture of phases with unknown distribution of elements between them. Results of these studies could significantly increase the scientific level of the paper and accordingly its interest for readers.

Author Response

(The authors gave the same response as above.)
